# Height Measurement for Meter Wave Polarimetric MIMO Radar with Electrically Long Dipole under Complex Terrain

**Yuwei Song and Guimei Zheng \***

Air and Missile Defense College, Air Force Engineering University, Xi'an 710051, China
**\*** Correspondence: zheng-gm@163.com

**Abstract:** Height measurement of meter wave radar is a difficult and important problem. This paper studies the height measurement of meter wave polarimetric (MWP)-MIMO array radar under complex terrain. The traditional electrically short dipole has low radiation efficiency, and the collocated dipole vector antenna has strong mutual coupling. This paper proposes to use electrically long dipoles and separated vector antennae to solve the problems of low radiation efficiency and strong mutual coupling. In addition, different from the traditional flat terrain, the research of this paper is based on the conditions of complex undulating terrain. First, the height measurement signal model of the MWP-MIMO radar with separated electrically long dipole under the complex terrain is derived. Then, a preprocessing method of block orthogonal matching pursuit is proposed to obtain the coarse estimation of the target's elevation. Then, under the guidance of the coarse estimation, the generalized MUSIC algorithm is used to obtain the high-precision elevation estimation of the target, and then the height measurement of the target is obtained according to the geometric relationship. Finally, the effectiveness of the proposed algorithm is proved by computer simulations.

**Keywords:** meter wave radar; MIMO Radar; height measurement; electrically long dipole; DOA estimation

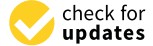



## 1. Introduction

The definition of meter wave (MW) radar is that the operating frequency is in the meter band [1,2], and the typical meter wave radar is VHF and UHF band radar. The development of the MW radar has a long history, which can be traced back to World War II, and plays an important role in air defense's early warning. However, the development of the MW radar is limited by the broad beam width, which leads to the beam sweeping over the ground during ground-to-air detection so that the amplitude of the ground-reflected echo is strong. Therefore, it encounters coherent multipath during low-altitude target detection, which makes the beam split and the fluctuating signal-to-noise ratio (SNR). Finally, the MW radar has poor elevation measurement accuracy, which cannot meet the guidance accuracy requirements. It is well known that the target's height is actually calculated by measuring the target elevation. Therefore, target angle estimation [3,4], especially low elevation target height measurement of the MW radar, has always been one of the important problems in array radar signal processing. At present, there are abundant research results on MW's radar height measurement with a conventional array. Among them, two types of methods are the focus of research: one is the transplantation and improvement of traditional super-resolution algorithms [5,6], in which the terrain factor is the focus [7–9]. The other is the applications based on artificial intelligence [10].

On the other hand, MIMO radar has the advantage of waveform diversity [11–13]. Applying the MIMO system to the MW radar [14], the study of DOA estimation of low-elevation targets in the MW-MIMO radar has attracted extensive attention. Considering the specular reflection of the smooth surface of flat terrain, unlike conventional array radar, where one target corresponds to two received paths, MIMO radar has four transmission

paths corresponding to one target: a pair of transmitted direct waves and received direct waves, a pair of transmitted direct waves and received reflect waves, a pair of transmitted reflect waves and received direct waves, and a pair of transmitted reflect waves and received reflect waves. MIMO radar will face not only the problem of coherent source incidence but also the problem of mutual penetration of guidance vectors, or what some experts and scholars call cone angle merger [14]. Therefore, the traditional DOA estimation of coherent source targets cannot be directly applied to the height measurement of MIMO array radar. For example, the super-resolution algorithm technology represented by multiple signal classification (MUSIC) cannot be directly applied to the height measurement of MIMO array radar. The generalized MUSIC algorithm does not need to be decoherent, which is an important means to solve such problems. In addition, it combines the direct wave and the reflected wave by taking advantage of the geometric relationship between the direct wave signal and the reflected wave signal through the pre-obtained prior information, such as the target distance and antenna height, so that the generalized MUSIC can complete the estimation of the target elevation only by conducting a one-dimensional angle search, which greatly reduces the amount of computation. The actual terrain is not necessarily flat. For the MW-MIMO radar in complex terrain, it is necessary to establish a ground fluctuation model, establish a signal model that can match the terrain, and develop its corresponding low elevation estimation method. Our team [15–19] transplanted the beam split, adaptive beamforming method, generalized MUSIC, and height measurement of MW-MIMO radar under complex field conditions.

In addition, a polarization-sensitive array (PSA) has the advantage of polarization diversity [20,21], so DOA estimation based on PSA radar has many excellent research results. The development of PSA has also overcome many difficulties, ranging from the strong mutual coupling of collocated electromagnetic vector sensors (EMVS) to the weak mutual coupling of separated EMVS [22,23], from low radiation efficiency of small EMVS to high radiation efficiency of large EMVS [24–29]. Therefore, it is of great practical significance to study DOA estimation of separated large EMVS because it is feasible in engineering.

This paper combines MW-MIMO radar with the latest separated electrically long dipole to study the height measurement of meter wave polarimetric MIMO radar (MWP-MIMO) [30,31] under complex position conditions. Compared with the previous work, this article is closer to the real application scenario. First, the signal model is derived, and then its generalized MUSIC algorithm is developed, but it involves a large amount of multi-dimensional search computation. The geometric relationship between the direct wave and the reflected wave can be used to reduce the search dimension, but the generalized MUSIC still involves the search of the entire airspace, and the computation is still large. The block orthogonal matching pursuit (BOMP) algorithm [32,33] is used to preprocess to obtain the coarse estimation so as to narrow the search range and reduce the computation. First, the received data of the polarimetric MIMO array radar is sparsely processed, and it is transformed into a signal model suitable for the BOMP algorithm. Then, the coarse angle estimation is obtained by coarse grid search, and then the beam width of the polarimetric MIMO radar is taken as the search range, with this as the initial value center. The advantage of such processing is that it cannot only guarantee the advantages of high precision estimation of generalized MUSIC [34] but also greatly reduce its computational complexity.

## 2. Signal Model

Assume that an MWP-MIMO radar array system with a collocated transmitter and receiver has a $N$ transmitted array element, the transmitted vector signal is equal to $\boldsymbol{\varphi}(t) \in \mathbb{C}^{N \times 1}$, and the transmitted signal is an orthogonal signal, which is an important difference between the MIMO radar and the traditional phased array radar. Then, there are $\int_0^{T_p} \boldsymbol{\varphi}(t)\boldsymbol{\varphi}(t)^H dt = \boldsymbol{I}_N$, where $\boldsymbol{I}_N$ is a unit array of size $N$, and $T_p$ is the width of the signal pulse. It implies that the transmitted signals are orthogonal to each other. The MWP-MIMO array is a uniform linear array. The received array uses a two-component cross dipole to separate

the vector sensor. Therefore, the system is called the MWP-MIMO radar system. Assuming that the MWP-MIMO radar system is in the undulating reflected terrain, it implies that the reflected terrain is not flat; the entire array radar is shown in Figure 1.

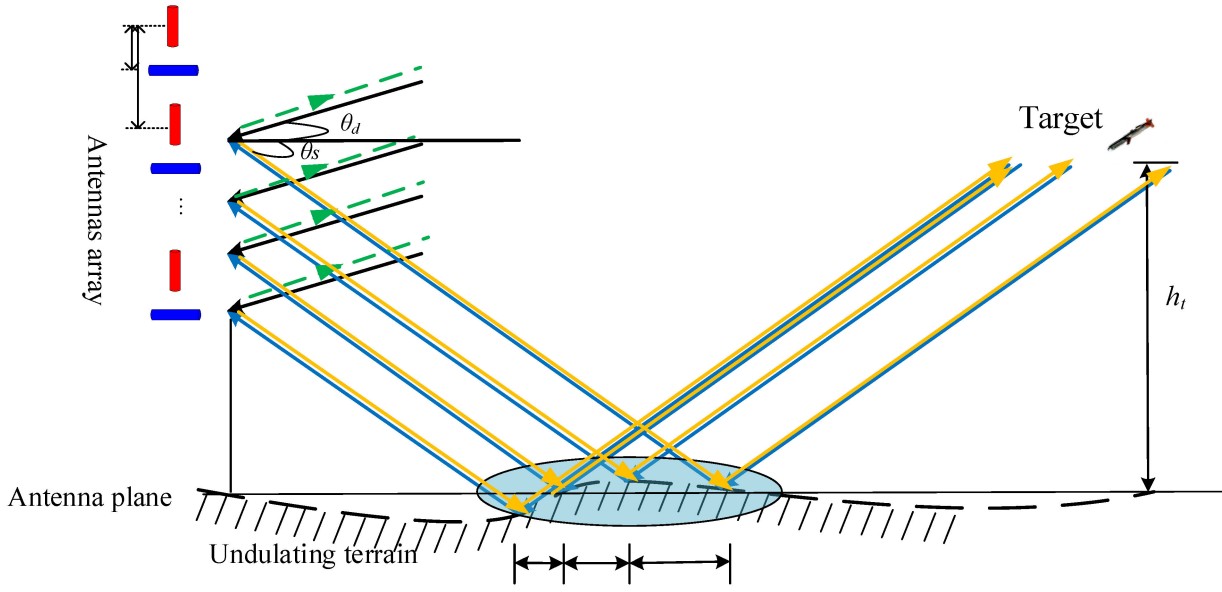

**Figure 1.** Height measurement signal model of separated electrically long dipole MWP-MIMO radar.

In Figure 1, $\theta_d$ is the direct wave angle, $\theta_s$ is the reflected wave angle, $D$ is the spacing between two vector antennae, and $d$ is the spacing between the two components in a single, separated vector antenna. $h_a$ is the antenna height (the lowest element position of the array antenna), $h_t$ is the target height, and $R$ is the horizontal distance between the point where the target is vertically projected to the ground and the radar antenna. Assuming that the transmitted signal of the radar is a horizontal polarization signal, and the vertical polarization antenna does not transmit the signal, the transmitted signals arriving at the target are equal [8]:

$$x(t) = [a_t(\theta_d) + e^{-j\delta} \rho_h \odot a_h(h_{s,i}) \odot a_t(\theta_s)]^{\mathrm{T}} \varphi(t) \tag{1}$$

$$\rho_h = [\rho_{h,1}, \rho_{h,2}, \cdots, \rho_{h,N}]^{T} \tag{2}$$

$$a_h(h_{s,i}) = [\exp(-j4\pi h_{s,1} \sin \theta_s / \lambda), \cdots, \exp(-j4\pi h_{s,N} \sin \theta_s / \lambda)]^{T} \tag{3}$$

where $a_t(\theta)$ is the steering vector of the transmitted array, and its value is equal to

$$a_t(\theta) = [1, \exp(-j2\pi D \sin(\theta)/\lambda), \cdots, \exp(-j2\pi(N-1)D \sin(\theta)/\lambda)]^{\mathrm{T}} \tag{4}$$

where $\theta$ can be $\theta_d$ or $\theta_s$. $\lambda$ is the wavelength. $\rho_h$ represents the reflection coefficient of each reflection point, and the reflection coefficient $\rho_h$ of a horizontal polarized wave can be calculated by Equation (5), and its value is as follows [32]:

$$\rho_h = \frac{\sin \theta_d - \sqrt{\varepsilon - \cos^2 \theta_d}}{\sin \theta_d + \sqrt{\varepsilon - \cos^2 \theta_d}} \tag{5}$$

where $\varepsilon$ is the complex permittivity of the surface, whose value can be expressed by the relative permittivity, $\varepsilon_r$, and the conductivity of the surface material, $\sigma_e$, is as follows.

$$\varepsilon = \varepsilon_r - j60\lambda\sigma_e \tag{6}$$

The $\varepsilon_r$ and $\sigma_e$ is determined by the medium of the reflection point, and the reflection point is a function of the elevation of the incident direct wave $\theta_d$ and the height of the reflection point $h_{s,i}$. The horizontal distance of the reflection point of each array element can be calculated by a simple trigonometric function as: $(h_i + h_{s,i}) \tan\theta_d$. $\delta = \frac{4\pi h_a h_t}{R\lambda}$ refers to the phase difference because of the electromagnetic wave path difference between the direct electromagnetic wave and the reflected electromagnetic wave. In meter wave radar, a range resolution unit is larger than the wave path difference; therefore, the direct and reflected waves cannot be distinguished in the range dimension, which causes signal reduction and beam split and lead to the difficulty of height measurement.

Each received vector sensor uses the vector form of a separated orthogonal two-component. Because it is assumed that the MIMO radar system has a common transceiver, the spatial received steering vector, $\boldsymbol{a}_r(\theta)$, is the same as the spatial transmitted steering vector, $\boldsymbol{a}_t(\theta)$, that is $\boldsymbol{a}_r(\theta) = \boldsymbol{a}_t(\theta)$. Then, the whole received signals of the MWP-MIMO radar are equal to

$$
\begin{aligned}
\boldsymbol{x}_1(t,\tau) &= \left[ \boldsymbol{a}_r(\theta_d) \otimes \boldsymbol{a}_d^L(\theta_d, \gamma, \eta) + \mathrm{e}^{-\mathrm{j}\delta}[(\boldsymbol{\rho}_h + \boldsymbol{\rho}_v) \odot \mathbf{a}_h(h_{s,i}) \odot \mathbf{a}_r(\theta_s)] \otimes \boldsymbol{a}_s^L(\theta_s, \gamma, \eta) \right] \\
&\quad \beta(\tau)x(t) + \boldsymbol{n}(t,\tau) \\
&= \left[ \boldsymbol{a}_r(\theta_d) \otimes \boldsymbol{a}_d^L(\theta_d, \gamma, \eta) + \mathrm{e}^{-\mathrm{j}\delta}[(\boldsymbol{\rho}_h + \boldsymbol{\rho}_v) \odot \mathbf{a}_h(h_{s,i}) \odot \mathbf{a}_r(\theta_s)] \otimes \boldsymbol{a}_s^L(\theta_s, \gamma, \eta) \right] \\
&\quad \beta(\tau)[\boldsymbol{a}_t(\theta_d) + \mathrm{e}^{-\mathrm{j}\delta}\boldsymbol{\rho}_h \odot \boldsymbol{a}_h(h_{s,i}) \odot \boldsymbol{a}_t(\theta_s)]\mathbf{T}\boldsymbol{\varphi}(t) + \boldsymbol{n}(t,\tau) \in \mathbb{C}^{2N \times 1}
\end{aligned}
\tag{7}
$$

$$
\boldsymbol{\rho}_v = [\rho_{v,1}, \rho_{v,2}, \cdots, \rho_{v,N}]^T
\tag{8}
$$

where $\beta(\tau) = \alpha \exp(\mathrm{j}2\pi f_d \tau)$ is the complex reflection coefficient between different pulses, which is an unknown complex constant. It is assumed that it follows the Swelling 2 distribution. $\rho_h$ is already defined in Equation (5), the vertical polarization reflection coefficient can be expressed as the following equation [35]:

$$
\rho_v = \frac{\varepsilon \sin\theta_d - \sqrt{\varepsilon - \cos^2\theta_d}}{\varepsilon \sin\theta_d + \sqrt{\varepsilon - \cos^2\theta_d}}
\tag{9}
$$

where

$$
\boldsymbol{a}_d^L(\theta_d, \gamma, \eta) = -[\boldsymbol{\Phi}(\theta_d)\boldsymbol{g}(\gamma, \eta)] \odot \underbrace{\begin{bmatrix} l_x(\theta_d) \\ l_y(\theta_d)\sec(\theta_d)e^{-j\frac{2\pi}{\lambda}d\sin\theta_d} \end{bmatrix}}_{\boldsymbol{u}(\theta_d)}
\tag{10}
$$

$$
\boldsymbol{a}_s^L(\theta_s, \gamma, \eta) = -[\boldsymbol{\Phi}(\theta_s)\boldsymbol{g}(\gamma, \eta)] \odot \underbrace{\begin{bmatrix} l_x(\theta_s) \\ l_y(\theta_s)\sec(\theta_s)e^{-j\frac{2\pi}{\lambda}d\sin\theta_s} \end{bmatrix}}_{\boldsymbol{u}(\theta_s)}
\tag{11}
$$

where $\boldsymbol{u}(\theta_s)$ indicates the influence of the electrically long dipole and the separated array element on the array response.

$$
\boldsymbol{g}(\gamma, \eta) \overset{\text{def}}{=} \begin{bmatrix} \sin\gamma e^{j\eta} \\ \cos\gamma \end{bmatrix}
\tag{12}
$$

where $\gamma$ is the auxiliary polarization angle and $\eta$ is the polarization phase difference.

$$
\boldsymbol{\Phi}(\theta_s) = \begin{bmatrix} -1 & 0 \\ 0 & \cos\theta \end{bmatrix}
\tag{13}
$$

$$
l_x(\theta) = -\frac{\lambda}{\pi} \frac{1 - \cos(\pi\frac{L}{\lambda})}{\sin(\pi\frac{L}{\lambda})}
\tag{14}
$$

$$
\ell_y(\theta) = -\frac{\lambda}{\pi} \frac{1}{\sin(\pi\frac{L}{\lambda})} \frac{\cos\left(\pi\frac{L}{\lambda}\sin(\theta)\right) - \cos(\pi\frac{L}{\lambda})}{\cos(\theta)}
\tag{15}
$$

where $\theta$ can be $\theta_d$, $\theta_s$. $L$ represents the length of the electrical dipole, and its range is $L \in [0.1\ 1]\lambda$. The length of an electrical dipole is comparatively longer, and its radiation efficiency is much higher than that of a short electric dipole. The transmitted signal vector $\boldsymbol{\varphi}(t)$ is used for match filtering the above equation and vectorizing the data; thus, we can obtain

$$
\begin{aligned}
\boldsymbol{x}(\tau) = &\left[ \boldsymbol{a}_r(\theta_d) \otimes \boldsymbol{a}_d^L(\theta_d, \gamma, \eta) + \mathrm{e}^{-\mathrm{j}\delta}[(\boldsymbol{\rho}_h + \boldsymbol{\rho}_v) \odot \boldsymbol{a}_h(h_{s,i}) \odot \boldsymbol{a}_r(\theta_s)] \otimes \boldsymbol{a}_s^L(\theta_s, \gamma, \eta) \right] \\
&\left[ \boldsymbol{a}_t(\theta_d) + \mathrm{e}{-}\mathrm{j}\delta \boldsymbol{\rho}_h \odot \boldsymbol{a}_h(h_{s,i}) \odot \boldsymbol{a}_t(\theta_s) \right] \mathrm{T}\beta(\tau) + \mathrm{vec}[\boldsymbol{N}(\tau)] \in \mathbb{C}^{2N^2 \times 1}
\end{aligned}
\tag{16}
$$

If the original noise is a zero mean Gaussian random process, then there is $E[\boldsymbol{n}(t_1, \tau)\boldsymbol{n}(t_2, \tau)^{\mathrm{H}}] = \sigma^2 \boldsymbol{I}_{2N}\omega(t_1 - t_2)$, where $\omega$ is the impact function and $\sigma^2$ is the power of noise. Then, the noise after matched filtering can be expressed as

$$
\bar{\boldsymbol{n}}(\tau) = \mathrm{vec}(\boldsymbol{N}(\tau)) = \mathrm{vec} \int_0^{T_p} \boldsymbol{n}(t, \tau)\boldsymbol{\varphi}(t)^{\mathrm{H}} dt
\tag{17}
$$

The noise characteristics after matched filtering are investigated by calculating the noise covariance matrix as follows.

$$
\begin{aligned}
\boldsymbol{R}_n = \quad & E\left\{ \mathrm{vec}[\boldsymbol{N}(\tau)]\mathrm{vec}[\boldsymbol{N}(\tau)]^H \right\} \\
= & E \int_0^{T_p} \int_0^{T_p} \boldsymbol{\varphi}(t_1)^* \otimes \boldsymbol{n}(t_1, \tau) \left\{ \boldsymbol{\varphi}^T(t_2) \otimes \boldsymbol{n}(t_2, \tau)^H \right\} dt_1 dt_2 \\
= & \int_0^{T_p} \int_0^{T_p} \boldsymbol{\varphi}(t_1)^* \boldsymbol{\varphi}^T(t_2) \otimes E \left\{ \boldsymbol{n}(t_1, \tau)\boldsymbol{n}(t_2, \tau)^H \right\} dt_1 dt_2 \\
= & \int_0^{T_p} \int_0^{T_p} \boldsymbol{\varphi}(t_1)^* \boldsymbol{\varphi}^T(t_2) \otimes \sigma^2 \boldsymbol{I}_{2N}\omega(t_1 - t_2) dt_1 dt_2 \\
= & \ \sigma^2 \boldsymbol{I}_N \otimes \boldsymbol{I}_{2N} = \sigma^2 \boldsymbol{I}_{2N^2}
\end{aligned}
\tag{18}
$$

It can be seen from the above formula that after match filtering with the original white noise, the covariance matrix of the noise is still a unit matrix, so the noise is still white noise.

### 3. Signal Model Preprocessing

First, we preprocess the received signal and convert Equation (16) into Equation (17).

$$
\begin{aligned}
\boldsymbol{x}(\tau) = \quad & [\boldsymbol{a}_t(\theta_d) + \mathrm{e}^{-\mathrm{j}\delta}\boldsymbol{\rho}_h \odot \boldsymbol{a}_h(h_{s,i}) \odot \boldsymbol{a}_t(\theta_s)] \\
\otimes & \left[ [\boldsymbol{a}_r(\theta_d) \otimes \boldsymbol{a}_d^L(\theta_d, \gamma, \eta) + \mathrm{e}^{-\mathrm{j}\delta}[(\boldsymbol{\rho}_h + \boldsymbol{\rho}_v) \odot \boldsymbol{a}_h(h_{s,i}) \odot \boldsymbol{a}_r(\theta_s)] \otimes \boldsymbol{a}_s^L(\theta_s, \gamma, \eta) ] \right] \\
& \cdot \bar{\beta}(\tau) + \boldsymbol{n}(\tau) \in \mathbb{C}^{2N^2 \times 1}
\end{aligned}
\tag{19}
$$

Equation (19) can be transformed using the Kronecker product to meet the combination rate and distribution rate into:

$$
\boldsymbol{x}(\tau) = \left[
\begin{array}{c}
\boldsymbol{a}_t(\theta_d) \otimes \boldsymbol{a}_r(\theta_d) \otimes \{[\boldsymbol{\Phi}(\theta_d)\boldsymbol{g}(\gamma, \eta)] \odot \boldsymbol{u}(\theta_d)\} \\
+ \mathrm{e}^{-\mathrm{j}\delta}[\boldsymbol{\rho}_h \odot \boldsymbol{a}_h(h_{s,i}) \odot \boldsymbol{a}_t(\theta_s)] \otimes \boldsymbol{a}_r(\theta_d) \otimes \{[\boldsymbol{\Phi}(\theta_d)\boldsymbol{g}(\gamma, \eta)] \odot \boldsymbol{u}(\theta_d)\} \\
+ \boldsymbol{a}_t(\theta_d) \otimes \mathrm{e}^{-\mathrm{j}\delta}[(\boldsymbol{\rho}_h + \boldsymbol{\rho}_v) \odot \boldsymbol{a}_h(h_{s,i}) \odot \boldsymbol{a}_r(\theta_s)] \otimes [[[\boldsymbol{\Phi}(\theta_s)\boldsymbol{g}(\gamma, \eta)] \odot \boldsymbol{u}(\theta_s)]] \\
+ \mathrm{e}^{-\mathrm{j}\delta}\boldsymbol{\rho}_h \odot \boldsymbol{a}_h(h_{s,i}) \odot \boldsymbol{a}_t(\theta_s) \otimes \mathrm{e}^{-\mathrm{j}\delta}[(\boldsymbol{\rho}_h + \boldsymbol{\rho}_v) \odot \boldsymbol{a}_h(h_{s,i}) \odot \boldsymbol{a}_r(\theta_s)] \otimes [[[\boldsymbol{\Phi}(\theta_s)\boldsymbol{g}(\gamma, \eta)] \odot \boldsymbol{u}(\theta_s)]]
\end{array}
\right]
\tag{20}
$$
$$
\cdot \bar{\beta}(\tau) + \boldsymbol{n}(\tau) \in \mathbb{C}^{2M^2 \times 1}
$$

Equation (20) is the incident signal received model of steering vector synthesis. When the steering vector and reflection coefficient are classified into one category, and the electromagnetic wave path difference and polarization parameters are classified into one category, Equation (20) can be further deformed to obtain:

$$x(\tau) = \begin{bmatrix} a_t(\theta_d) \otimes a_r(\theta_d) \otimes \mathbf{\Pi}(\theta_d) \\ a_t(\theta_d) \otimes [(\rho_h + \rho_v) \odot a_h(h_{s,i}) \odot a_r(\theta_s)] \otimes \mathbf{\Pi}(\theta_s) \\ a_t(\theta_s) \otimes a_r(\theta_d) \otimes \mathbf{\Pi}(\theta_d) \\ a_t(\theta_s) \otimes [(\rho_h + \rho_v) \odot a_h(h_{s,i}) \odot a_r(\theta_s)] \otimes \mathbf{\Pi}(\theta_s) \end{bmatrix} \begin{bmatrix} I_2 \\ e^{-j\delta}I_2 \\ e^{-j\delta}I_2 \\ e^{j-2\delta}I_2 \end{bmatrix} g(\gamma,\eta)\beta(\tau) + \bar{n}(\tau) \tag{21}$$

where $\mathbf{\Pi}(\theta_d) = \mathbf{\Phi}(\theta_d) \odot [u(\theta_d)u(\theta_d)]$, $\mathbf{\Pi}(\theta_s) = \mathbf{\Phi}(\theta_s) \odot [u(\theta_s)u(\theta_s)]$. The reflection coefficient $\rho_h, \rho_v$ and wave path difference $\delta$ are functions of $\theta_d, \theta_s$; therefore, the steering vector for the generalized MUSIC algorithm is defined as follows.

$$A_{\text{gmusic}}(\theta_d, \theta_s, h_{s,i}, \rho_h, \rho_v) = \begin{bmatrix} a_t(\theta_d) \otimes a_r(\theta_d) \otimes \mathbf{\Pi}(\theta_d) \\ a_t(\theta_d) \otimes [(\rho_h + \rho_v) \odot a_h(h_{s,i}) \odot a_r(\theta_s)] \otimes \mathbf{\Pi}(\theta_s) \\ a_t(\theta_s) \otimes a_r(\theta_d) \otimes \mathbf{\Pi}(\theta_d) \\ a_t(\theta_s) \otimes [(\rho_h + \rho_v) \odot a_h(h_{s,i}) \odot a_r(\theta_s)] \otimes \mathbf{\Pi}(\theta_s) \end{bmatrix} \in \mathbb{C}^{2N^2 \times 8} \tag{22}$$

$$\bar{\beta}(\gamma, \eta, \tau) = \begin{bmatrix} I_2 \\ e^{-j\delta}I_2 \\ e^{-j\delta}I_2 \\ e^{j-2\delta}I_2 \end{bmatrix} g(\gamma,\eta)\beta(\tau) \in \mathbb{C}^{8 \times 1} \tag{23}$$

According to the above definition, Equation (19) can be abbreviated as follows:

$$x(\tau) = A_{\text{gmusic}}(\theta_d, \theta_s, h_{s,i}, \rho_h, \rho_v)\bar{\beta}(\gamma, \eta, \tau) + \bar{n}(\tau) \tag{24}$$

In Equation (24), $\bar{\beta}(\gamma, \eta, \tau)$ can be equivalent to eight coherent incident sources, that is, to define a new signal source.

## 4. Combination Algorithm of Fine and Coarse Estimation

Firstly, the single snapshot BOMP algorithm is used to obtain the coarse estimation of the target elevation, and then the coarse estimation is used to delimit a small angle search area. Then, the generalized MUSIC algorithm without decoherence is used to obtain the precise estimation of the target elevation, and finally, the target height is obtained. According to this analysis logic, we will introduce the BOMP coarse estimation algorithm, angle narrowing range determination, and generalized MUSIC algorithm.

### 4.1. Coarse Height Measurement of MWP-MIMO Radar Based on BOMP

Next, we introduce a method that can obtain coarse elevation estimation under a single snapshot to narrow the search range so as to reduce the amount of calculation. First, making $\tau = 1$ implies that the snapshot is only one in Equation (24). The steering vector and the received signal source are sparsely represented. It is necessary to construct a two-dimensional matrix to construct a sparse over the complete matrix $A(\psi), \psi = (\theta_d, \theta_s)$, which requires a lot of computation. Here, the mathematical relationship in Equation (25) between the direct and reflected wave can also be applied to cut down the dimension of the over-complete dictionary, as follows.

$$\theta_s = -\arctan(\tan\theta_d + 2h_a/R) \tag{25}$$

That is $\psi = (\theta_d)$, $J$ represents the length of the dictionary, which $J$ is far greater than the number of targets. Here, we set the number of targets as $K$ (in fact, for the height measurement model, the target is one). The above equation can be written as

$$x = A(\psi)z + n \tag{26}$$

where $z$ represents the $K$ block sparse vector. The length of the block is equal to eight. Equation (26) can use the BOMP algorithm to find the support position so as to calculate

the elevation angle. The following is a brief description of the relationship between the BOMP algorithm and Equation (26).

First of all, $(\theta_{d,k})_{k=1}^{K}$ is divided into a grid, $J$, in which $J \gg K$. This $(\theta_{d,l})_{l=1}^{J}$ corresponds to the block sparse vector $z \triangleq \left[ \left( z^{[1]} \right)^{T}, \cdots, \left( z^{[l]} \right)^{T}, \cdots, \left( z^{[L]} \right)^{T} \right]^{T} \in \mathbb{C}^{8J \times 1}$, where $z^{[l]} = [z_1^{[l]}, z_2^{[l]}, \cdots, z_8^{[l]}]^{T} \in \mathbb{C}^{8 \times 1}$. If the target falls on the search grid (as for the case where the target falls between two grids, it is a coarse estimate and does not affect the final result), then

$$\forall 1 \le k \le K, \exists 1 \le l \le J, \text{s.t.} z^{[l]} = y^{[k]} \tag{27}$$

Therefore, $\forall 1 \le l \le J$, and there are only two values $z^{[l]}$:

$$\begin{cases} z_i^{[l]} = 0 & (1 \le i \le 8) \\ z_i^{[l]} \ne 0 & (1 \le i \le 8) \end{cases} \tag{28}$$

The target is transformed into finding the support position of the target $K$ in the sparse vector $z$ of the block, and the support position is the elevation position in the sparse dictionary of the target. Now the optimization problem is transformed into

$$\begin{cases} \underset{z}{\arg\min} \; \|z\|_{B,0} \\ \\ \text{s. t. } A_a(\psi)z = r - \hat{n} \end{cases} \tag{29}$$

where $\|z\|_{B,0} \triangleq \# \left\{ l \, \middle| \, z^{[l]} \ne \boldsymbol{0}, 1 \le l \le J \right\}$, $\hat{n}$ is the estimated value of $n$, and the estimated value can be obtained by covariance matrix decomposition [9]. The objective optimization function of Equation (29) is to find the sparsest solution (therefore, $l_0$ norm). The constraint of Equation (29) is that the sparse solution is required to meet the observation conditions (that is, it is consistent with the collected data). When the number of snapshots of the received data is one, the above equation is a standard $K$ block sparse problem. Its support position can be calculated using the BOMP algorithm, and the result is recorded as $\hat{z}^{[k]}, k = 1, \cdots, K$. Then, it is easy to calculate the target position, $\theta_k^{\text{coarse}}$. For the sake of illustration, the concept of support position is given here, which refers to the non-zero position of the target in the sparse dictionary. In this paper, it is the target position, namely

$$\hat{\theta}_k^{\text{coarse}} = V \cdot (\hat{z}^{[k]} - 1)/8, k = 1, \cdots, K \tag{30}$$

wherein $V$ represents the interval degree in the construction of a sparse dictionary. When the number of snapshots is large, the results obtained by the above method can be averaged. In fact, the results here are only used as the results of coarse estimation. In order to reduce the heavy computational complexity, the single-snapshot BOMP algorithm can be directly used.

### 4.2. Search Range of Height Measurement Search Algorithm for MWP-MIMO Radar

The initial elevation estimation of MWP MIMO radar has been obtained by the above section. This is the center of the search algorithm to determine the search scope. The basic principles of search scope are given below. First, define the search range $\theta = (\theta^{\text{coarse}} + \theta_1, \theta^{\text{coarse}} - \theta_1)$, and the core is to determine the value $\theta_1$. The criterion here is defined as half beam width, where the search range is equal to one beam width. If the target power drops beyond the beam width, then the target cannot be detected, so it is reasonable to search within a single beam width. The beam width of the MIMO radar is $\theta_{mb} = \frac{50.7\lambda}{ND \cos \theta_d}, \theta_1 = \frac{\theta_{mb}}{2}$, where $M$ represents the number of equivalent aperture array elements of the MIMO radar, which is equal to $M = 2N - 1$. For the MIMO radar with

a common transceiver, the number of effective aperture array elements can also be easily calculated for the non-equidistant linear array, so it will not be repeated here.

### 4.3. Generalized MUSIC of MWP-MIMO Radar in Complex Terrain

The traditional MUSIC method is based on the orthogonality between the noise subspace and the signal subspace after the covariance matrix decomposition. Specifically, we calculate the covariance matrix of the received data in Equation (24), decompose it into eigenvalues, and then select the corresponding eigenvectors outside the two largest eigenvalues to form a noise subspace $E_n$. For the conventional MUSIC algorithm, the steering vector is used to project the noise subspace to obtain the spatial spectrum of MUSIC, and then the angle is extracted. However, for MWP-MIMO radar, the direct wave steering vector $a_t(\theta_d)$ and the reflected wave steering vector $a_r(\theta_s)$ intersect with each other, so the spatial smoothing algorithm has failed. However, the algorithm using generalized MUSIC does not have this problem. It directly uses the orthogonal relationship between the noise subspace, $E_n$, and signal subspace, and the signal subspace has the same space as the steering matrix, $A_{\mathrm{gmusic}}$. Therefore, the noise subspace, $E_n$, and the steering matrix, $A_{\mathrm{gmusic}}$, are orthogonal, and the spatial spectrum of the generalized MUSIC is obtained, as shown in Equation (31).

$$P(\theta_d) = \frac{\det\left[A_{\mathrm{gmusic}}{}^H(\theta_d, \theta_s, h_{s,i}, \rho_h, \rho_v) A_{\mathrm{gmusic}}(\theta_d, \theta_s, h_{s,i}, \rho_h, \rho_v)\right]}{\det\left[A_{\mathrm{gmusic}}{}^H(\theta_d, \theta_s, h_{s,i}, \rho_h, \rho_v) E_n E_n{}^H A_{\mathrm{gmusic}}(\theta_d, \theta_s, h_{s,i}, \rho_h, \rho_v)\right]} \tag{31}$$

The above equation involves multi-dimensional search, in which the height and coefficient of the reflection point, $h_{s,i}, \rho_h, \rho_v$, are considered to be known, and the dimension of the reflected wave and the direct wave is reduced by using a geometric relationship so that the one-dimensional search angle value can be obtained, and the final target height value can be calculated.

## 5. Proposed Algorithm Steps

According to the above analysis, the steps of the height measurement method for MWP-MIMO radar in complex terrain are as follows.

Step 1: Use BOMP to obtain the initial estimation and determine the search scope. The calculation flow of the BOMP algorithm is given in Table 1.

**Table 1.** Calculation process of BOMP algorithm.

| | |
|---|---|
| Input | Vectorized data, number of angle grids, and number of targets after matched filtering |
| Initialize | Initialize the residual with the received data, initialize the dictionary with the number of angle grids, and initialize the support set |
| Iteration | 1. Use residual and dictionary to calculate projection;<br>2. Find the maximum coordinate value of the block according to the projection, and put this value coordinate into the block support set;<br>3. Use block support set to update residuals;<br>4. Iteration of Steps 1 to 3. Stop when the number of iterations reaches the target number; |
| Output | Calculate the block support vector using the block support set |

Step 2: Use the initial estimation value obtained in Step 1 to determine the angle search range, then use the generalized MUSIC to obtain the final angle estimation value, and then convert it into the target height value according to the target distance.

## 6. Computer Simulation Results

*Example 1* of support position recovery of BOMP pretreatment.

The number of transmitted vector antennas of the MWP-MIMO radar is equal to $N = 4$. The radar operating frequency is equal to 300 MHz; then, the wavelength is equal to $\lambda = 1$ m. Therefore, it is a meter wave radar. The separation vector antenna spacing is half wavelength, $d = \lambda/2$, and the spacing between vector elements is one wavelength, $D = \lambda$. The direct wave angle of the target is set to 4 degrees, and the reflected wave angle is computed by using Equation (25). Polarization auxiliary angle is $\gamma = 45°$ and polarization phase difference in polarization parameters is $\eta = 90°$, then the signal is left-hand circular polarization signal. The antenna height is $h_a = 5$ m, and the target height is $h_t = 7000$ m. The number of snapshots is set to 10 and the SNR = 5 dB. For the convenience of calculation, the reflection coefficient of the dielectric surface is set to the same freshwater scene, in which the dielectric constant $\varepsilon_r = 80$ and surface material conductivity is $\sigma_e = 0.2$. The height of the target undulating reflection point is set to $h_{s,i} = [0.3\ 0.6\ 2.1\ 3.2]$ meters. $[: n° :]$ means that each interval $n°$ takes a numerical value to construct a sparse dictionary. Here, in $n = 1$, it is set that the first support position of the theoretical elevation value of the target is $4 \times 8 + 1 = 33$. From the above analysis, for the BOMP algorithm, we know that its block is eight, so there must be eight consecutive points during sparse recovery for one target. For this simulation, because the target is at 4 degrees and the dictionary interval is 1 degree, theoretically, the support position of the target should be between 33 and 40. In fact, we can see that the simulation results are indeed in this position, which is consistent with the theoretical analysis and verifies the correctness of the algorithm in this paper.

Figure 2 shows the support position estimation results of one of the independent experiments. It can be seen that the BOMP algorithm can correctly estimate the target elevation.

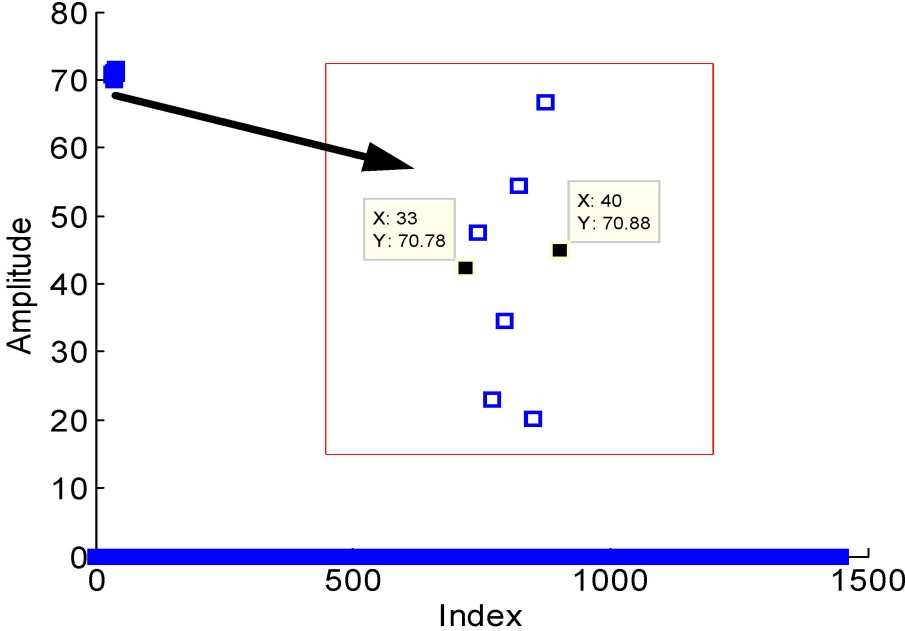

**Figure 2.** Support position estimation results based on BOMP sparse recovery.

*Example 2* of the verification of correctness of search range selection.

The simulation conditions are the same as in Example 1. In this example, we add the generalized MUSIC after the BOMP algorithm. Figure 3 shows the first estimation result of the full space search of the generalized MUSIC. The beam width calculated by the formula $\theta_{mb} = \frac{50.7\lambda}{ND\cos\theta_d}$ is $\theta_{mb} \approx 6.35°$. Figure 4 shows the search range results of 10 independent experiments based on the generalized MUSIC after BOMP preprocessing. It can be seen

from the figure that the algorithm in this paper can correctly narrow the search range, thus reducing the amount of computation.

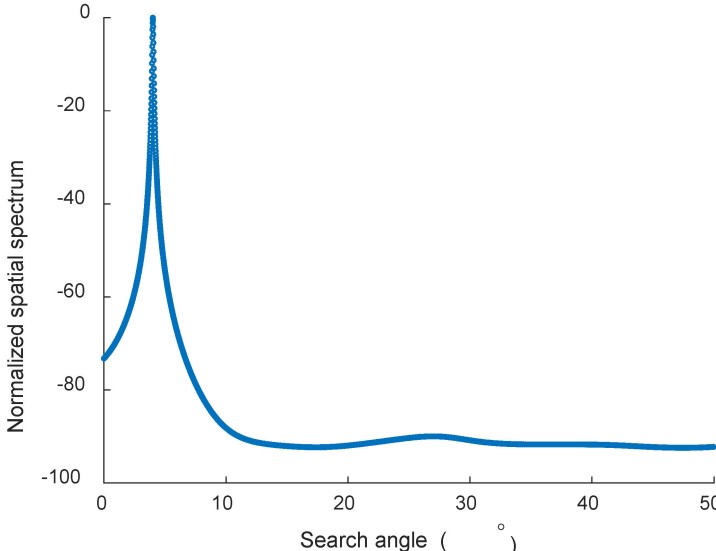

**Figure 3.** Estimation results of generalized MUSIC full space search.

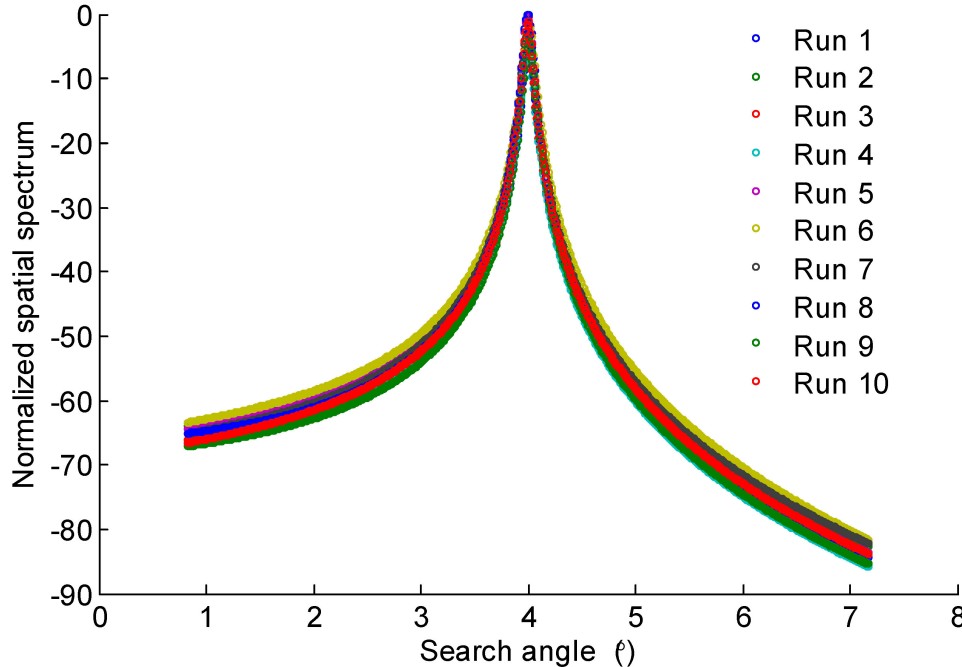

**Figure 4.** Search range results of 10 independent experiments based on generalized MUSIC after BOMP preprocessing.

*Example 3* of the verification of the proposed algorithm in complex terrain.

In this simulation, we only change the terrain information and the angle of the target; the other simulation conditions are the same as in Example 1. To verify the correctness of the proposed algorithm in complex and changeable terrains. The direct wave angle of the target is set at 5 degrees. The reflection coefficient of the dielectric surface is set as very dry ground, in which the dielectric constant $\varepsilon_r = 4$ and surface material conductivity is $\sigma_e = 0.006$. The height of the target undulating reflection point is set to $h_{s,i} = [1.3\ 0.7\ 1.1\ 2.2]$ meters. Figure 5 shows the support positions of sparse recovery. From Figure 5, we can see that the theoretical values are consistent with the simulation

results, which equals 41 to 48. Figure 6 shows the spatial spectrum of an experimental generalized MUSIC algorithm, which can correctly obtain the target angle value.

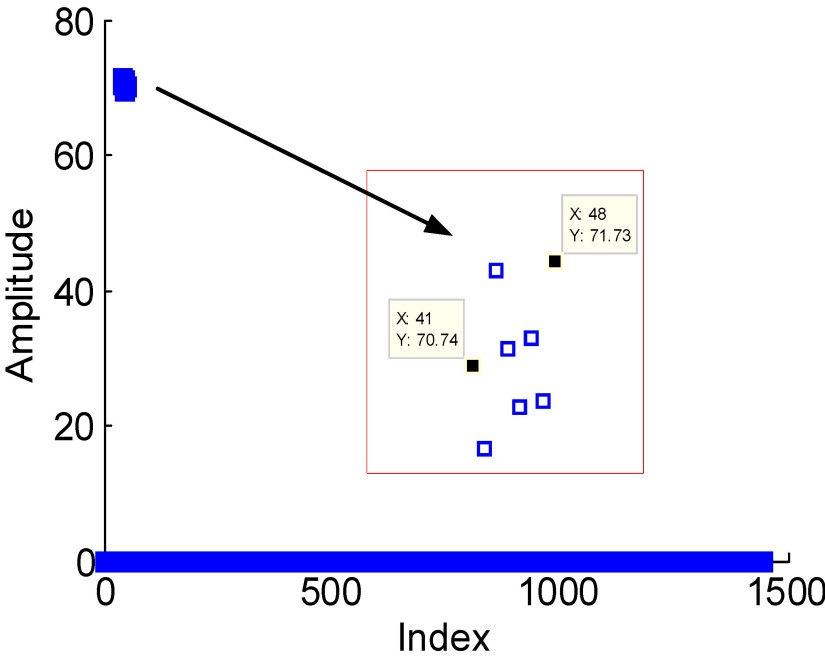

**Figure 5.** Support position estimation results based on BOMP sparse recovery with the 5 degrees of target's angle.

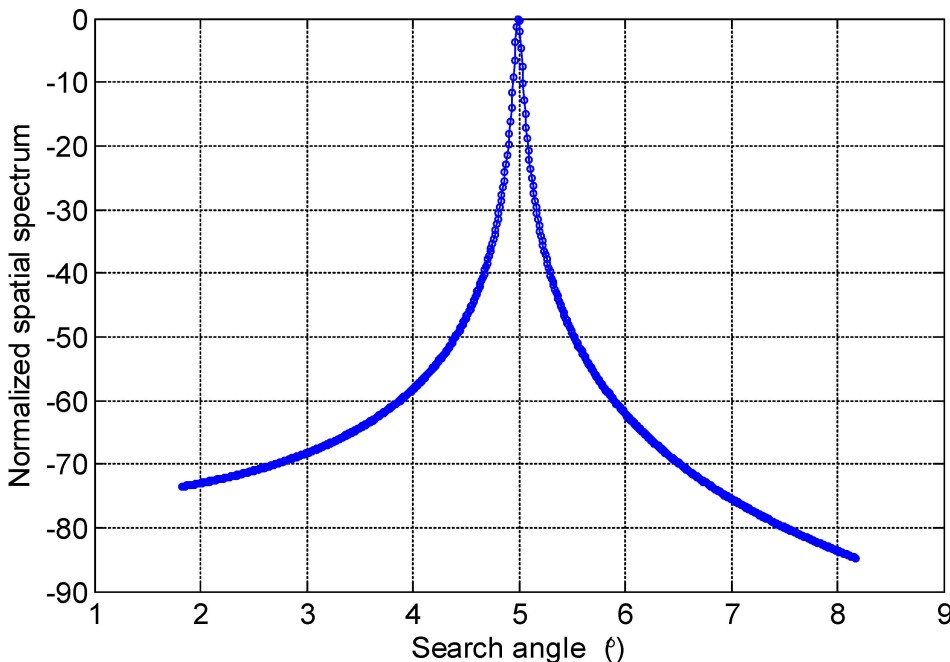

**Figure 6.** Search range results based on generalized MUSIC after BOMP preprocessing.

*Example 4* of the elevation and altitude estimation results of the generalized MUSIC algorithm. The simulation conditions are the same as in Example 1. In this example, the SNR changes from 0 to 20. It completes 1000 Monte Carlo experiments and defines the root-mean-square error (RMSE) to measure the estimation performance as

$\text{RMSE} = \sqrt{\frac{1}{\text{Monte}} \sum_{p=1}^{\text{Monte}} E[(\hat{\alpha}_p - \alpha_p)^2]}$, where $\hat{\alpha} = \hat{\theta}_d$ is the direct wave angle estimate and $\alpha = \theta_d$ is the real value of the target angle. When the target height is estimated to be $\hat{\alpha} = \hat{h}_t$, then $\alpha = h_t$ is the true value of the target height. Figures 7 and 8 show the curve of RMSE variation with SNR for angle estimation based on the generalized MUSIC algorithm after BOMP preprocessing and the curve of height measurement error variation with SNR. The proposed algorithm with the terrain matching method is called Match BOMP-G-MUSIC in all figures, and the terrain mismatch method is called Mis-match BOMP-G-MUSIC in all figures. It can be seen from Figures 7 and 8 that the height measurement estimation of polarimetric MIMO radar is correctly obtained by the algorithm of terrain matching in this paper, while the estimation error of the mismatch method is large and belongs to biased estimation.

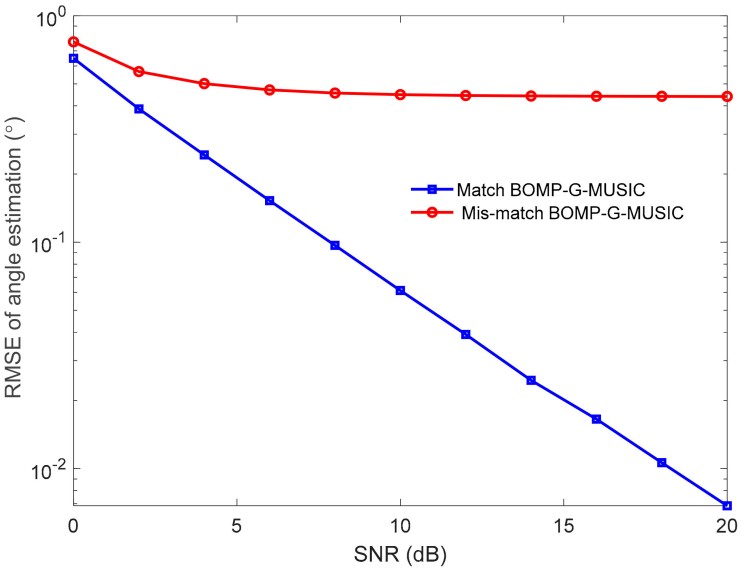

**Figure 7.** Low elevation estimation RMSE for terrain mismatch and matching.

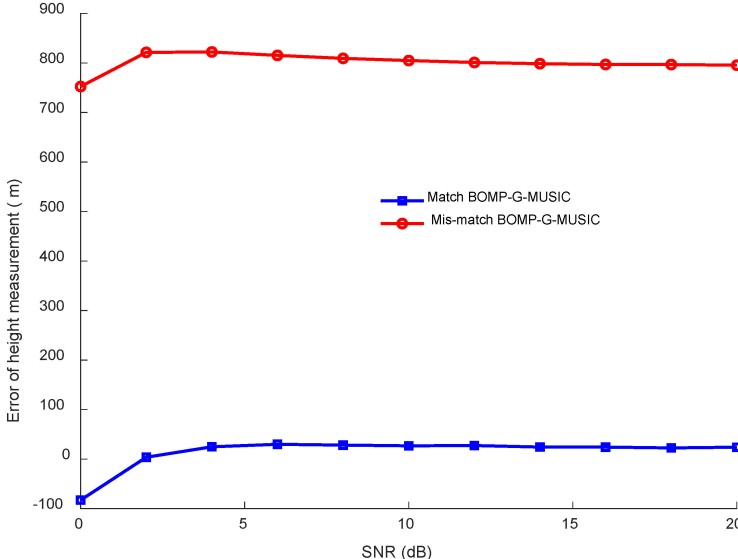

**Figure 8.** Height estimation error of terrain mismatch and matching.

*Example 5* of the comparison of operation time between preprocessing and non-preprocessing algorithms.

The simulation conditions are the same as in Example 1. In this example, it completes one Monte Carlo experiment. Figure 9 shows the comparison between the operation time of the full space search of the generalized MUSIC and the operation time of the generalized MUSIC after the BOMP preprocessing. It can be seen from shows the comparison between the operation time of the full space search that the algorithm in this paper can effectively reduce the amount of calculation. The more the number of array elements, the more obvious the advantages of the algorithm in this paper.

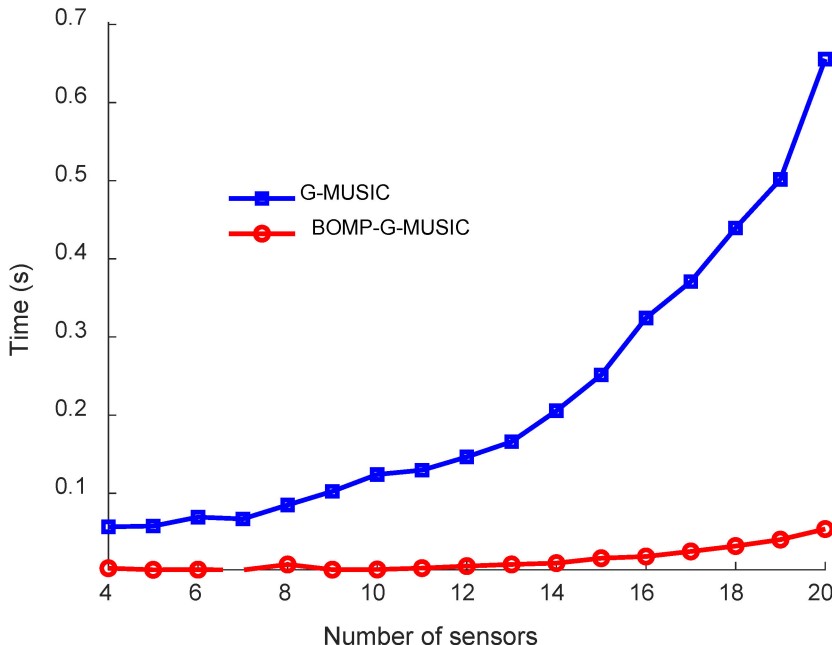

**Figure 9.** Comparison results of operation time of preprocessing and non-preprocessing algorithms.

In addition, the height measurement accuracy of our proposed algorithm ultimately depends on the generalized MUSIC algorithm, so using the BOMP algorithm will not increase or decrease the height measurement accuracy but only reduce the calculation amount. So we did not compare their estimation accuracy because the estimation accuracy is the same.

## 7. Discussion

In this section, we discuss the highlight and concretion of our proposed algorithm. For coherent signal sources, this paper uses a combination of fine and coarse estimation to obtain the height of the target. It should be noted here that both the BOMP algorithm and the generalized MUSIC algorithm do not require decoherence.

The main difference between our paper and existing papers is the analyses as follows. The polarization steering vectors of the two papers are consistent, except that one vector antenna is six-dimensional and the other is two-dimensional. However, their spatial steering vectors have essential differences. One terrain is undulating, and the other is flat. It is because of this that precisely makes it more difficult for us to measure the height of the target. Obviously, the complexity is higher because of the need for terrain information. We can compare the signal models of the two papers, specifically Equation (16) in this manuscript and Equation (10) in reference [31]. We can see the differences between them. We talk about them in detail. The spatial steering vectors equal $(\boldsymbol{\rho}_h + \boldsymbol{\rho}_v) \odot \boldsymbol{a}_h(h_{s,i}) \odot \boldsymbol{a}_r(\theta_s)$ and the $\boldsymbol{a}_r(\theta_s)$, respectively. It is obvious that the spatial steering vector of the undulating terrain in this paper is much more complex than that of the flat terrain. It involves the inhomogeneity of the terrain and the reflection medium, which makes it difficult to solve



mathematically. It needs to do some preprocessing, which is also one of the difficulties of this manuscript.

The disadvantage of our proposed algorithm is that it needs to accurately grasp the terrain of the ground and the reflection coefficient of the ground, which sometimes takes a lot of time to measure. In addition, although the preprocessing of the algorithm greatly reduces the amount of computation, the search algorithm may not be able to complete the real-time processing of radar measurement. These are the factors that limit the practical application of the proposed algorithm.

## 8. Conclusions

This paper studies the problem of low elevation measurement of MWP-MIMO radar based on BOMP preprocessing. The work performed in this paper is based on the complex terrain, combined with the advantages of polarization information and waveform diversity. In addition, the separated electromagnetic vector structure and electrically long dipole are also used. It not only solves the problem of strong mutual coupling but also improves radiation efficiency. It can be said that it is a relatively advanced array structure mode at present, especially from the perspective of a polarization array. In addition, the advanced BOMP algorithm is also applied to reduce the amount of computation. The non-decoherent sparse recovery algorithm BOMP is proposed to coarsely measure the low elevation angle of the target and then narrow the angle search range of the traditional generalized MUSIC algorithm, thus reducing the computational complexity. In fact, BOMP can also be directly used without considering the amount of calculation. The simulation results show that the surface terrain matching algorithm is better than the terrain mismatch estimation algorithm.

**Author Contributions:** Conceptualization, Y.S. and G.Z.; methodology, G.Z.; software, Y.S.; writing—original draft preparation, Y.S.; writing—review and editing, G.Z. All authors have read and agreed to the published version of the manuscript.

**Funding:** This work was supported by the National Natural Science Foundation of China under grant 61971438.

**Data Availability Statement:** The MATLAB codes can be obtained upon reasonable request by sending an email to zheng-gm@163.com.

**Conflicts of Interest:** The authors declare that there are no conflict of interest.

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
