# Peer review of "Height Measurement for Meter Wave Polarimetric MIMO Radar with Electrically Long Dipole under Complex Terrain"

_remotesensing, doi:10.3390/rs15051265_

Round 1

Reviewer 1 Report

This manuscript proposes to use electrically long dipoles and separated vector antenna to solve the problems of low radiation efficiency and strong mutual coupling. BOMP is introduced to obtain the coarse estimation of the target’s elevation, reducing the complexity of the calculation. The simulation results show that using the surface terrain matching algorithm is less inaccurate than the terrain mismatch estimation algorithm for measuring height. However, the manuscript has a high degree of similarity to articles Height Measurement with Meter Wave Polarimetric MIMO Radar: Signal Model and MUSIC-like Algorithm and BOMP-based angle estimation with polarimetric MIMO radar with spatially spread crossed-dipole in terms of model building, lack of innovation and low quality graphs.

Therefore, I don't recommend this manuscript for publication in the ‘Remote sensing’. Please find the specific comments below.

Specific comments:

1,BOMP is used to perform pre-processing and thus reduce the amount of calculations. However, the manuscript only provides a comparative analysis in terms of time and lacks a comparative analysis for accuracy before and after pre-processing, making it difficult to demonstrate the advantages of BOMP.

2,The introduction of BOMP into MUSIC to reduce computational effort has been mentioned in some articles, where is the innovation in this manuscript?

3,Only one Table exists for the manuscript, where is the Table 2 in Proposed algorithm steps?

4,The Signal mode and Signal model preprocessing sections of the manuscript are highly similar to those of Height Measurement with Meter Wave Polarimetric MIMO Radar: Signal Model and MUSIC-like Algorithm derivation. The process is highly similar and the authors need to point out the differences and innovations in these two sections.

5, The quality of the article diagrams needs to be improved. Further explanation is needed for figure 2, what do dots 33 & 40 mean and why does this occur? What do the different lines in Figure 4 mean? This needs to be labelled by the author. Why is there no value for the BOMP-G-MUSIC line segment in Fig. 7 at 7 sensors.

6, Formula 18 exists a problem.

Author Response

Dear Editor, Dear Reviewers,

This is a revised manuscript. The original manuscript ID: remotesensing-2194582. Thank you very much for the reviewers’ insightful comments and suggestions. It is valuable. We have revised the manuscript according to the reviewers’ and the editor’s comments. Our point-to-point response is in attachment. The revised contents are marked with RED font in the revised manuscript.

Best wishes

Guimei Zheng

Reviewer 2 Report

The manuscript demonstrates the continuation of a series of studies regarding the development of MIMO radar measurements. It presents a new processing algorithm for undulating terrain. The theoretical part in the first part of the manuscript is considered in sufficient detail, but the section on computer simulation could be better presented in more detail. For example, for approbation, the parameters of water as a reflective surface, as well as a small set of height of the target undulating reflection points, are used. As a result, it does not clearly illustrate the problem of complex terrain. I suppose that the addition of several computer models made it possible to better prove the effectiveness of the proposed approach and the closeness of the shown studies to real scenarios. It would also be useful to see a short ‘Discussion’ section at the end, where the authors would discuss the strengths and weaknesses of the proposed approach circumstantially and compare with the algorithms from references [17,19,32].

Author Response

Dear Editor, Dear Reviewers,

This is a revised manuscript. The original manuscript ID: remotesensing-2194582. Thank you very much for the reviewers’ insightful comments and suggestions. It is valuable. We have revised the manuscript according to the reviewers’ and the editor’s comments. Our point-to-point response is attached below. The revised contents are marked with RED font in the revised manuscript.

Best wishes.

Guimei Zheng

Round 2

Reviewer 1 Report

The manuscript has been modified in response to the comments, but there are still some problems. As a result, I recommend a Minor Revision before it can be accepted.

Specific comments:

1, The signal model preprocessing using kronecker product to meet the combination rate and distribution rate, the accuracy needs to be analyzed. Why was the kronecker chosen for the manuscript?

2, The methodology limitation should be mentioned.

Author Response

Dear Editor, Dear Reviewers,

This is a revised manuscript. The original manuscript ID: remotesensing-2194582. Thank you very much for the reviewers’ insightful comments and suggestions. It is valuable. We have revised the manuscript according to the reviewers’ and the editor’s comments. Our point-to-point response is attached below. The revised contents are marked with RED font in the revised manuscript.

Reviewer 2 Report

The authors have done significant work to improve the manuscript and expand the experiment. An important discussion section has been added. However, in my opinion, it would be correct to add to this section a mention of the limitations of the technique and some assumptions about what conditions it will not be able to work correctly. Should units be specified for surface material conductivity?

Author Response

Dear Editor, Dear Reviewers,

This is a revised manuscript. The original manuscript ID: remotesensing-2194582. Thank you very much for the reviewers’ insightful comments and suggestions. It is valuable. We have revised the manuscript according to the reviewers’ and the editor’s comments. Our point-to-point response is in attachment. The revised contents are marked with RED font in the revised manuscript.
